# Youth Water Polo Performance Determinants: The INEX Study

**DOI:** 10.3390/ijerph19094938

**Published:** 2022-04-19

**Authors:** Francisco Noronha, Sofia Canossa, João P. Vilas-Boas, José Afonso, Flávio Castro, Ricardo J. Fernandes

**Affiliations:** 1Centre of Research, Education, Innovation and Intervention in Sport, Faculty of Sport, University of Porto, 4200-450 Porto, Portugal; up201302164@fade.up.pt (F.N.); scanossa@fade.up.pt (S.C.); jpvb@fade.up.pt (J.P.V.-B.); jneves@fade.up.pt (J.A.); 2Porto Biomechanics Laboratory, Faculty of Sport, University of Porto, 4200-450 Porto, Portugal; 3Aquatic Sport Research Group, Universidade Federal do Rio Grande do Sul, Porto Alegre 90690-200, Brazil; souza.castro@ufrgs.br

**Keywords:** age group, water polo, anthropometry, motor performance, development

## Abstract

Due to the growing engagement of youth in water polo practice, we aimed to characterize age-grouped players across anthropometric, general and specific motor abilities and contextual domains. We have also examined the associations of players’ specific skills with their anthropometric and general motor characteristics. One-hundred-and-one male water polo players, grouped into 12-, 13- and 14-year age cohorts were recruited. One-way ANOVA explained age-cohort variance, and a multiple linear regression was used to assess the association between variables. The variance in cohorts was explained by arm span (25%), stature, hand breadth and length (17%) fat-free mass (18%), 20 m sprint (16%), sit-ups (18%), medicine ball throw (27%), anaerobic (31%) and aerobic performance (21%), change of direction (18%), and in-water vertical jump (14%). The variance of in-water vertical jump, 10 m sprint, change of direction and aerobic fitness for players’ anthropometric characteristics were, 32, 25, 14 and 10% (respectively). The players’ upper-limb explosive power explained 30, 22 and 17% of variance for in-water vertical jump, 10 m sprint and aerobic fitness, respectively. Body mass had an inverse, and arm span had a direct association with in-water vertical jump and swim velocity capability, arm span had an inverse and direct association with change of direction and aerobic fitness, respectively. The upper limbs’ explosive power related directly to in-water vertical jump and aerobic fitness skills, but inversely with 10 m sprint scores.

## 1. Introduction

Revealing the determinants underlying talent identification and promotion aiming for sports excellence has been a goal of researchers and coaches [1,2]. Studying the individual performance characteristics can show compensating phenomenon effects when assuming that they singly do not determine a player performance level but the combination among them does. Even if the number of studies focusing on water polo anthropometrics, general motor performance [3,4] and biomechanical [5,6] and physiological determinants [7,8] has increased over the last years, the players’ initial development processes are complex and require deeper theoretical and practical knowledge. This will allow for improvement of the training process and, consequently, the game’s fundamental structures [4,9,10].

It is well known that water polo players’ anthropometric features are highly related with high performance levels, influencing intensive offensive and defensive actions in each playing position [11,12]. Highly competitive performance requires adequate general motor skills and physiologic responses positively influencing water polo game success [13,14,15]. Consequently, youth’s formative years should enhance the overall game demands by including combined exercises such as HIIT and strength training concurrently with specific water polo training [16,17] and promoting the associations between numerous intense bursts of activity, intermittent efforts and specific playing positions. These will help maximize the young water polo players’ developmental program [18,19].

Training specific motor abilities is essential due to the demanding, specialized proficiencies required for high-quality performance during the early development stages [20,21], with its underdevelopment being associated with late sports performance [1,5]. Water polo players are constantly changing their body position during the game, and their aptitude depends on a specific conditioning level (e.g., the out-of-the-water body elevations are required in all specific game motor abilities [22,23]). Therefore, detailing the upper- and lower-limb-specific techniques can provide helpful information regarding water polo propulsion efficiency [24,25,26]; these scientific observations complement intuitive judgments about young talents based on the elite level requirements and, with success, standards of longitudinal personal profiles [2,22]

In addition, the context quality in which players are inserted has a close relationship with their psychological well-being and motivation, with coaches as the direct and responsible elements for youth players in their formative years, based on general and specific knowledge of their motor development [27,28]. This knowledge goes far beyond the participation in formal coach-education programs and has to be supported by scientific evidence [29]. Children’s participation in sport modifies their behaviour, cognition and parental affection through contextual social interaction [30], and family involvement has to be of an ideal level, which has motivational implications for young players over time [31,32].

Since studies related to youth water polo players’ general and specific motor abilities have been scarce [33], we have implemented a multivariate approach to youth water polo performance by characterizing age-group players across anthropometrics, general motor performance, specific motor abilities and contextual domains. Complementarily, we have examined the potential association of the assessed specific motor abilities with anthropometric characteristics and general motor performance scores. We hypothesized that older age-group players would display higher body size values and better general/specific motor performance than their younger peers, and that anthropometric and general motor performance variables would be directly associated with specific motor abilities.

## 2. Materials and Methods

One-hundred-and-one age-group water polo players, their coaches and parents from the research project “In search of excellence: a mixed-longitudinal study in young athletes (INEX) [34]” participated in the current study. Players were 13 ± 0.61 ((12–14) as minimum-maximum) years old, had over two years of competitive practice and an average of four training sessions (6 h, with 30 min dryland physical exercise plus 60 min in-water specific exercise) per week. To characterize players’ anthropometrics, general motor performance, specific motor abilities and contextual characteristics, a multilevel system model dividing players into age cohorts 1, 2 and 3 (12.00–12.99, 13.00–13.99 and 14.00–14.99 years old, respectively) was used. All measurements were conducted by trained researchers, and written informed consent was obtained from parents/legal guardians and players, with the study being approved by the local university Ethics Committee (CEFADE 13.2017).

Regarding the anthropometric features, stature was measured using a Harpenden stadiometer (Holtain Ltd., Crymych, UK, precision to the nearest millimetre), arm span and hand breadth and length were assessed with an anthropometer and a sliding calliper (Siber-Hegner, GPM, Zurich, Switzerland), and body mass, body fat and fat-free mass were obtained using a Tanita BC-418 MA bioimpedance scale with a 1.4% CV [35]. Biological maturation was evaluated with a prediction equation based on age, sex, stature, sitting stature and body mass, allowing for the assessment of the peak stature velocity and timing of positive and negative maturity offset values, evidencing the years that the player is beyond or before the peak stature velocity age [36]. The anthropometric technical error of measurement was 0.8 cm for stature, 0.9 cm for arm span, 0.1 cm for hand breadth and length and 0.9 and 0.3 kg for body mass and fat-free mass. All measurements followed the International Working Group on Kinanthropometry protocols [37].

For evaluating players’ general motor performance, the following were assessed: (i) aerobic fitness using the Yo-Yo intermittent endurance test (level 1) [38]; (ii) middle body strength through the abdominal and hip flexor sit-up endurance test (number of repetitions during 60 s) [39]; (iii) the static handgrip strength test using a hand-held digital dynamometer (T.K.K.5401 Grip-D, Takei, Japan, accuracy of ±2.0 kg) [40]; (iv) lower-limb explosive power conducting a squat and a countermovement jump on a 2000 Hz AMTI OR6-WP force platform with 1.8–4.2% CV [41] (Advanced Mechanical Technology Inc., Watertown, MA, USA); (v) upper-limb explosive power by throwing a 3 kg medicine ball [42]; (vi) running velocity performing a 20 m sprint using a photoelectric cell system Speed Trap II (Brower timing systems LLC., Draper, UT, USA) with 2.3% CV [39]; and (vii) the change of direction and body control using the *t*-test using the above-referred photoelectric cell system [43]. Two trials were performed for sit-ups, sprint and *t*-test, and three trials were conducted for vertical jumps, with the best value used for posterior analysis. Two trials for each hand at the handgrip test and three medicine ball throw trials were performed, with the mean value being accepted for data analysis. The technical error of measurement was 26.9 m for the Yo-Yo test, 0.7 repetitions for sit-ups, 0.9 kg for handgrip, 0.0 and 0.1 s for 5 and 20 m sprints, 0.1 s for *t*-test and 0.6 cm for squat/countermovement jumps.

Water polo players’ physiological characterization was conducted by assessing their aerobic fitness using the multistage shuttle swim test (10 m water polo front-crawl trials, starting at 0.9 m·s^−1^ and increasing to 0.05 m·s^−1^ per level [44]) and their anaerobic fitness, employing the 10 m sprint test (considering the best time from three water polo front-crawl trials). The specific motor abilities were also evaluated based on methodologies described in the specialized literature, with the jump skill being assessed using the in-water vertical jump test (with players starting from the basic water polo position and touching their hand to the screen as high as possible; Figure 1, left panel) [20]. Two trials were conducted, with the highest jump (observed using a Sony Handycam HDR-PJ530 digital video camera [20], resolution of 1920 × 1080 p) being considered for analysis [9]. The technical error of measurement was 0.9 cm for in-water vertical jump, 0.007 m/s for aerobic fitness and 0.1 s for 10 m sprint.

In addition, the change of direction was evaluated by an agility test with the player inside a square removing the ball as fast as possible inside the arches next to the players who previously received the ball (Figure 1, right panel) [45] with 0.1 s as a technical error of measurement. The shot effectiveness and velocity were measured by the shot-on-goal test in the presence of a goalkeeper (Figure 2) [46]. For evaluating shot accuracy, using a canvas with eight holes attached to the goal (Figure 2) [46], each player performed six trials from the penalty mark after previous displacement (three upper-limb dribbling cycles) and six other throws without displacement (for both above-referred shot types). Trials were registered by one researcher using a digital camera (Sony Handycam HDR-PJ530, Tokyo, Japan); specific categorization described elsewhere [46]. Shot percentage precision, efficacy and accuracy were calculated using a mathematical formula [15,47], and shot velocity was evaluated through a 10 Hz frequency radar (Inc., Flat Salkerpro, TX, USA) placed 8 m from goal with a 0.045 m·s^−1^ sensitivity. The technical error of measurement was 2.5 and 2.6% for direct and after-dribble shot, and 1.8 and 1.7% for direct and after-dribble precision shot, respectively.

The contextual evaluation focused on both players, coaches, parents and clubs. Water polo coaches’ demographic data, professional experience and academic and sport education was assessed. In addition, coaches’ competencies were also assessed using the coaching efficacy scale [28] and the leadership scale for sport [48]. The players’ parents’ assessment was centered on their perceptions of beliefs, support and parental encouragement. Lastly, information regarding the clubs’ characteristics, infrastructure, human resources and communication was collected.

### Statistical Analyses

Data homogeneity and normality were tested using Levene and Shapiro–Wilk tests, with descriptive statistics presented as means and standard deviations. A one-way ANOVA compared age cohorts, and the Bonferroni test was used for post-hoc multiple comparisons. Partial eta squared (η^2^) measures explained variance, i.e., the effect size, interpreted as 0.01 small, 0.06 moderate and 0.14 large. A multiple linear regression was used to assess the association of the anthropometrics, general motor performance and specific motor abilities in the studied players’ cohorts. Data analyses were performed using IBM SPSS Statistics 27.0 (Chicago, IL, USA), and a 5% significance level was accepted.

## 3. Results

### 3.1. Anthropometry

In Table 1 the selected anthropometric characteristics of the studied age-group water polo players can be observed. The older players were heavier, more mature and had more fat-free mass than their younger counterparts. In addition, players of cohort 2 were taller and exhibited greater arm span and hand breadth and length compared to their peers. Body fat scores were similar between groups.

### 3.2. General Motor Performance

Table 2 displays players’ general motor performance test results. Older players outperformed their peers in middle-body and handgrip strength and upper-limb explosive power and were more agile and aerobically fit. Players of cohort 2 achieved superior squat and countermovement jump values and were faster in the 5 and 20 m runs).

### 3.3. Specific Motor Abilities

In Table 3, the selected specific motor ability scores of the studied age-group water polo players can be seen. The older players performed faster changes of direction, jumped higher in water and presented better specific aerobic fitness. Players of cohort 2 swam faster and reached better shot efficacy after dribbling. However, direct precision shot and precision shot after dribble percentages were similar between cohorts.

### 3.4. Contextual Domain

Coaches were former water polo players with 10.13 ± 6.37 [3,4,5,6,7,8,9,10,11,12,13,14,15,16,17,18,19,20,21,22,23,24] years of coaching experience, and 20% (*n* = 3) were coaching national federation and/or regional association teams. In terms of education, 27% (*n* = 4) had a coaching level three diploma, 33% (*n* = 5) were level one and two coaches, and only one had a level four certificate. The majority revealed higher levels of perceived coach efficacy, and four coaches displayed knowledge of character-building and teaching. The coaching behaviour subscales reported a moderate level for positive feedback, training and instruction. Complementarily, players’ parents showed a low-to-moderate level of support and encouragement and reported spending 4 h/week supporting their children’s sports activities. A significant variation amongst the water polo clubs was observed, specifically regarding the number of players (58.14 ± 36.03), historical background of a water polo department (22.57 ± 17.68 years) and national and regional competition wins (8.86 ± 13.50 and 4.29 ± 7.54, respectively). Clubs presented a similar number of competitive levels (4 ± 1.98), they did not own the facilities where water polo players train and compete, and 57% (*n* = 4) had gym. Social media was a communication tool commonly used by all clubs, and one also used a radio station or TV/online channel for this purpose.

### 3.5. Specific Motor Ability Associations

The association between age-group water polo players’ specific motor abilities, their anthropometrics and general motor performance characteristics are provided in Table 4. The adjusted R^2^ for in-water vertical jump, 10 m sprint, change of direction and aerobic fitness for players’ anthropometric characteristics were 32, 25, 14 and 10%, respectively (all for *p* < 0.01). The players’ upper-limb explosive power explained 30, 22 and 17% of variance for in-water vertical jump, 10 m sprint and specific aerobic fitness, respectively (all for *p* < 0.01). The beta coefficients for the selected variables showed that body mass had an inverse, and arm span had a direct association with in-water vertical jump and swim velocity capability. In addition, arm span had an inverse and direct association with change of direction and aerobic fitness, respectively. Upper-limb explosive power related directly to in-water vertical jump and aerobic fitness skills, but inversely with 10 m sprint scores.

## 4. Discussion

The knowledge of youth motor development and formative years in sport using a multivariate approach will improve players’ developmental program process and fundamental game structures. Contrary to what was hypothesised, water polo players of cohort 2 presented better anthropometric characteristics in most of the studied variables, except those regarding body mass and fat-free mass, for which the older players displayed higher values, probably due to the biological maturity influence on size and adiposity [1,49]. Older players had superior general and specific motor performance, excluding lower-limb strength, run and swim velocities, for which they were outperformed by cohort 2. As expected, body mass, arm span and upper-limb explosive power values were moderately associated with specific motor abilities, even if not observed for shooting skills.

Water polo players’ anthropometric profile is relevant to their individual performance but also to team’s game achievements [11]. In previous studies, Italian water polo players of similar age to our cohort 1 and 2 participants showed identical stature and superior body mass [50], and Italian elite players presented higher stature, body mass and arm span compared to our cohort 2 adolescents [33]. Similarly, the current study’s cohort 3 subjects evidenced lower values in the above-referred variables compared to Serbian players [51]. These results agree with previous data reporting greater physical dimensions in older and elite water polo players compared to amateur peers [12], differences that could influence water polo general and specific motor performance skills.

High levels of strength, muscular power, endurance and velocity execution of specific abilities enhances match performance due to presence of body contact and fast swimming and offence and defence actions that are typical during the game [3]. Time frame for strength acquisition influences the development of youth players’ performance, consequently, specific motor abilities’ effectiveness [50]. However, the youth players’ growth process and learning rhythm can influence several motor performance capacities. In fact, in a previous study, players of the same age as cohort 1 had higher lower-limb strength compared to the same and greater upper-limb strength than our sample [18]. In addition, all players of the current study had lower middle-body strength compared to 13-year-old sub-elite and elite water polo players [33], which can be justified by the range of maturation within each cohort group.

Water polo players move their bodies vertically out of the water to execute specific motor abilities such as the pass, shot and block or to prevent the opponent from doing these [20]. Thus, players’ over-the-water surface body elevation is crucial for success of specific technical abilities and should be based on effective eggbeater technique [6,24]. It was reported in 12-year-old and 14- year-old water polo players’ equal and inferior in-water vertical jump values (respectively; [2,18]) compared with our data, which can be explained by their superior anthropometric features, with physical performance influencing the results. Our cohort 3 players outperformed their peers in the vertical jump ability, probably due to their knowledge about its positive transfer to other technical abilities that are vital to success at the superior competition level where they play [5,13], or because they are at a stage where power is more developed. In fact, water polo players are in the vertical position 47% of the match time performing activities with intermittent intensities [22].

It is known that specific motor abilities’ execution deteriorates along a water polo match due to physical fatigue [23,52], reinforcing the importance of maintaining adequate levels of physiological conditioning to meet technical training and competition demands [7]. To perform high-intensity activities interspersed by short recovery periods, it is extremely important to have an adequate energy provision both from aerobic and anaerobic metabolism [53]. Our study evidenced a better aerobic capability of the older players, probably linked to their higher training status, more matches played and superior age-group competition level. In fact, the need for higher aerobic energy supply [10,17] is well supported by the specific tactical situations inherent to the water polo game [8,19,44]. However, cohort 2 players displayed higher anaerobic capability scores when performing the head-above-water crawl technique, possibly because their higher lower-limb strength helped them maintain a higher horizontal body position to better overcome hydrodynamic drag [25]. Our younger players performed the anaerobic test in less than 10 s, which is in agreement with previous data pointing out that 33% of swimming bouts last than 10 s during youth water polo games [54].

The coaches’ knowledge and intervention in training and competition should be applied not only as a technical or tactical competence but in an integrative way. Coach education courses and their formal programs are associated with acquiring a predetermined amount of information to be a certified coach [55], with the current study’s water polo coaches having levels one and two of a certification level that has four levels. Their education and the fact that all were former players are consistent with the literature that highlights that the main source of reference for coaches during their first years of professional practice is their previous playing experience [29]. Complementarily, parental involvement plays a key role in the family support regarding the young water polo players’ participation and learning [27,31], with the current data showing a low-to-moderate level of parental support and encouragement in contrast to a strong level of assistance reported in a previous study [2].

Regarding the associations between the studied variables, we observed that arm span, body mass and upper-limb explosive power were the most strongly associated with specific motor ability performance. Cohort 2 players had higher arm span values, change of direction ability and shot percentages, in line with previous studies indicating the influence of arm span on specific match tactics and technical abilities [11,12]. Furthermore, it has been reported that body mass and body fat values are inversely related with power- and velocity-specific motor abilities [4,15], corroborating our results for sprint swimming and in-water vertical jump values. These specific abilities were expected to be strongly associated with upper-limb strength (as previously reported [18]), but the poor associations observed in our sample may possibly be explained by players’ poorer technical execution instead of power [20]. A study of elite players’ physiological profiles did not detected an association between strength and aerobic capability [16], in line with our finding of a weak association between selected strength variables and aerobic capability.

The observed differences between age groups in the various domains of the current study bring relevant knowledge to apply in the training process planning during youth’s formative years. Thus, an individualized training based on growth and development of young players is highly suggested. Even if the current study included a multivariate approach to youth water polo performance, it has the limitation of presenting a different number of players per cohort (small sample size for cohort 3). We therefore suggest future studies using a consistent sample per age group, increasing the number of participants, analysing female water polo players as well, and, if possible, conducting a longitudinal data analysis. In addition, future studies should consider focusing on how technical execution level influences specific motor ability performance, as well as the effects of maturation status on the age-group cohorts.

## 5. Conclusions

Data from the current study contribute to an increased knowledge on a multivariate approach of youth team players in general and adolescent water polo players in particular. Our findings indicate that anthropometric and general motor performance characteristics were associated with the performance of specific motor abilities that are essential for match success and players’ general development over the formative years. The current data provide an understanding of the general and specific player development process, which should be considered by coaches of youth players to improve players’ sports skills as a result of developing better training programmes.

## Figures and Tables

**Figure 1 ijerph-19-04938-f001:**
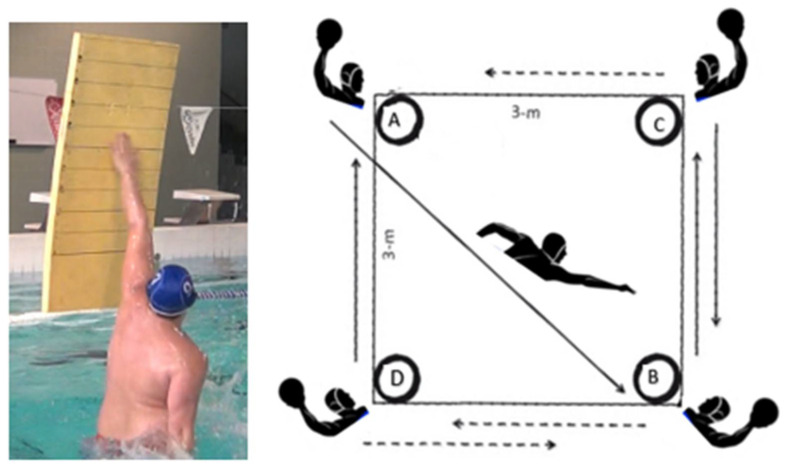
In-water vertical jump and agility tests (**left** and **right** panels, respectively).

**Figure 2 ijerph-19-04938-f002:**
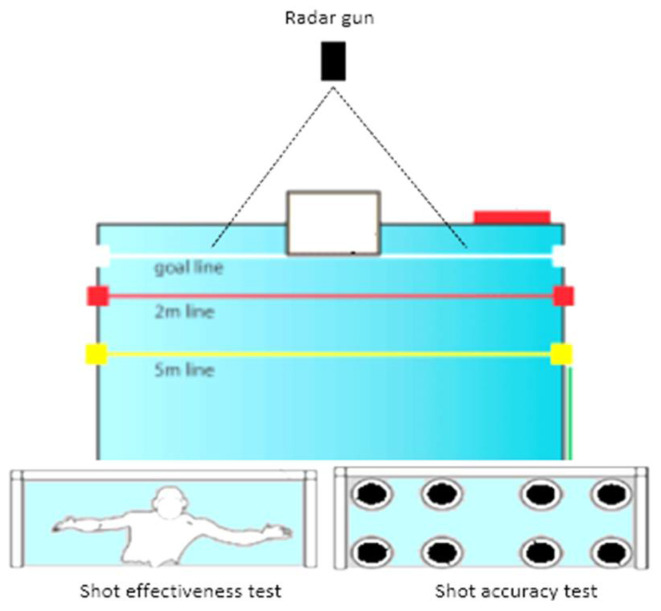
Shooting tests to goal and toward canvas (left and right panels, respectively).

**Table 1 ijerph-19-04938-t001:** Mean ± SD values of age-group players’ anthropometrics.

Variables	Cohort 1	Cohort 2	Cohort 3		η^2^
	(*n* = 34)	(*n* = 56)	(*n* = 11)	F (*p*)	
	Mean ± SD	Mean ± SD	Mean ± SD		
Age (years)	12.53 ± 0.28	13.50 ± 0.26	14.15 ± 0.16	220.255 ^†^	0.81
Stature (cm)	156.11 ± 7.28 **	163.47 ± 8.22	162.90 ± 3.77	10.00 ^†^	0.17
Body mass (kg)	49.86 ± 10.11	53.40 ± 8.55	57.94 ± 6.22	4.25 ^‖^	0.08
Arm span (cm)	158.80 ± 5.58 *	168.31 ± 8.60	165.40 ± 4.29	15.64 ^†^	0.25
Hand breadth (cm)	17.02 ± 1.13 **	18.02 ± 0.99	17.87 ± 0.65	10.09 ^†^	0.17
Hand length (cm)	19.01 ± 1.64 **	20.42 ± 1.51	20.31 ± 1.03	9.87 ^†^	0.17
Body Fat (kg)	10.33 ± 4.48	9.59 ± 1.10	10.21 ± 3.10	0.46 ^‖^	0.64
Fat-free mass (kg)	38.42 ± 6.54 **	44.32 ± 6.78	46.78 ± 4.64	10.79 ^†^	0.18
Offset maturation (years)	–1.13 ± 0.66 **	–0.14 ± 0.64	0.32 ± 0.41	35.27 ^†^	0.42

* and **: differences between cohort 2 and 3 (respectively). † and ‖: *p* < 0.001 and < 0.05 (respectively).

**Table 2 ijerph-19-04938-t002:** Mean ± SD values of age-group-general motor performance.

Variables	Cohort 1	Cohort 2	Cohort 3		η^2^
	(*n* = 34)	(*n* = 56)	(*n* = 11)	F (*p*)	
	Mean ± SD	Mean ± SD	Mean ± SD		
Handgrip (kg)	23.37 ± 6.28 **	28.60 ± 7.30	30.97 ± 10.08	6.90	0.13
5 m sprint (s)	1.38 ± 0.11	1.23 ± 0.20	1.32 ± 0.23	3.66 ^‖^	0.07
20 m sprint (s)	3.87 ± 0.31 *	3.60 ± 0.30	3.73 ± 0.17	8.75 ^†^	0.16
Sit-ups (reps)	32.19 ± 4.66 **	37.44 ± 5.88	38.36 ± 5.95	10.19 ^†^	0.18
Squat jump (cm)	21.25 ± 4.58 *	24.41 ± 5.35	22.86 ± 3.31	4.38 ^‖^	0.08
Countermovement jump (cm)	22.93 ± 4.78 *	26.15 ± 4.75	23.11 ± 2.07	5.89	0.11
Medicine ball throw (m)	2.95 ± 0.43 **	3.53 ± 0.55	3.78 ± 0.44	17.73 ^†^	0.27
*t*-test (s)	11.46 ± 0.74 **	10.94 ± 0.82	10.45 ± 0.92	7.79 ^†^	0.14
Yo-Yo IR1 (m)	336.97 ± 212.14 **	496.73 ± 263.05	603.64 ± 320.73	6.18	0.11

* and **: differences between cohort 2 and 3 (respectively). † and ‖: *p* < 0.001 and < 0.05 (respectively).

**Table 3 ijerph-19-04938-t003:** Mean ± SD values of age-group-specific motor abilities.

Variables	Cohort 1	Cohort 2	Cohort 3		η^2^
	(*n* = 34)	(*n* = 56)	(*n* = 11)	F (*p*)	
	Mean ± SD	Mean ± SD	Mean ± SD		
In-water vertical jump (cm)	110.26 ± 9.44 **	117.62 ± 8.41	118.00 ± 9.60	7.86 ^†^	0.14
Change of direction (s)	5.59 ± 0.65 **	4.96 ± 0.73	4.64 ± 0.90	10.30 ^†^	0.18
Direct shot (%)	50.49 ± 23.02	52.38 ± 26.48	60.60 ± 22.70	0.69	0.01
Shot after dribble (%)	43.14 ± 28.75 *	57.29 ± 17.83	45.45 ± 15.07	4.38 ^‖^	0.09
Direct precision shot (%)	20.59 ± 18.83	23.90 ± 18.83	16.67 ± 0.00	0.97	0.02
Precision shot after dribble (%)	22.06 ± 14.05	22.92 ± 14.05	21.21 ± 13.10	0.06	0.00
Aerobic fitness (m/s)	0.92 ± 0.03 **	0.96 ± 0.05	1.00 ± 0.08	11.91 ^†^	0.21
10 m sprint (s)	8.01 ± 0.86 **	6.97 ± 0.54	7.10 ± 1.03	20.94 ^†^	0.31

* and **: differences between cohort 2 and 3 (respectively). † and ‖: *p* < 0.001 and < 0.05 (respectively).

**Table 4 ijerph-19-04938-t004:** Association of water polo players’ specific motor abilities with their anthropometric and general motor performance characteristics.

Variables	Equation (Unstandardized Coefficients)
In-water vertical jump	y = 3.486 + (−0.256 body mass) + (0.759 arm span)
	y = 84.271 + (9.211 upper-limb explosive power)
10 m sprint	y = 16.496 + (−0.064 arm span) + (0.026 body mass)
	y = 9.757 + (−0.711 upper-limb explosive power)
Change of direction	y = 10.924 + (−0.035 arm span)
Aerobic fitness	y = 0.581 + (0.002 arm span)
	y = 0.828 + (0.037 upper-limb explosive power)

## Data Availability

Data presented in this study are available on request from the corresponding author. The data are not publicly available due to ethical reasons.

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
