# Peer review of "Youth Water Polo Performance Determinants: The INEX Study"

_ijerph, 2022, doi:10.3390/ijerph19094938_

Round 1

Reviewer 1 Report

I find your article interesting. The summary is appropriate and concise.

Correct introduction, which could mention studies on biological age , such as: Mirwald, et al. (2002), An assessment of maturity from anthropometric measurements. Medicine and Science in Sports and Exercise; and also: Gómez-Campos, et al. (2013), Assessment of biological maturation: uses and applications in the school environment, Revista Andaluza de Medicina del Deporte. 

Regarding the physical fitness assessment, have the authors considered the ALPHA Battery?

The methodological design is accurate. 

Clear results.

Discussion justifies the findings and correctly addresses the concerns of other researches as well as those of water polo coaches.

Author Response

Point 1: Correct introduction, which could mention studies on biological age , such as: Mirwald, et al. (2002), An assessment of maturity from anthropometric measurements. Medicine and Science in Sports and Exercise; and also: Gómez-Campos, et al. (2013), Assessment of biological maturation: uses and applications in the school environment, Revista Andaluza de Medicina del Deporte.

Response 1: We consider your suggestion and the first study was added in the methodology section.

Point 2: Regarding the physical fitness assessment, have the authors considered the ALPHA Battery?

Response 2: We didn’t use only one battery, all the tests selected were based on previous studies and also in some batteries as for example AAHPER. All the tests are properly referenced.

Point 3: The methodological design is accurate.

Clear results.

Discussion justifies the findings and correctly addresses the concerns of other researches as well as those of water polo coaches.

Response 3: Thank you for the comment.

Reviewer 2 Report

In my opinion, the abstract includes to much data. I would advise that you focus on the main results.

A second advice is on the discussion part: Please add some more information on biolocial age and matuation. Water polo is an athletic sport and this topic seem to be related. Therefore I advise to discuss this in this section. Maybe this can be a topic for further investigations.

Author Response

We would like to acknowledge for the time expended on reviewing our manuscript.

Point 1: In my opinion, the abstract includes too much data. I would advise that you focus on the main results.

Response 1: We understand you point of view, but we strongly believe that all the information present in the abstract are relevant and the number of words were respected.

Point 2: A second advice is on the discussion part: Please add some more information on biological age and maturation. Water polo is an athletic sport and this topic seem to be related. Therefore, I advise to discuss this in this section. Maybe this can be a topic for further investigations.

Response 2: Thank you for the comment. The cohorts were divide based on the competitive groups that are related with their chronological age and were also as a characterization point of view. The models were created using all the subjects together. We agree that this is an important issue and was included in discussion for further investigations.

Reviewer 3 Report

Youth water polo performance determinants: The INEX study

First of all, the reviewer would like to thank the authors for their work and efforts in trying to improve sports science knowledge. The study is well designed and well-written, with a great introduction proposing the usefulness of the topic and a clear outline of the research question. I suggest that the author modify/include some suggestions in order to improve the manuscript prior to be published:

Methods section

Line 83: The authors should add number of the Ethics file number

Table 1-2-3: The authors should add descriptor of pÆž2  such as small, moderate etc.

Results section

Overall the results are well-written and showed

Discussion section

Overall the discussion is well-written and incorporates relevant literature.

Tables and Figures

These sections are well designed and well-written.

Author Response

We would like to acknowledge for the time expended on reviewing our manuscript.

Point 1: Line 83: The authors should add number of the Ethics file number

Table 1-2-3: The authors should add descriptor of pÆž2  such as small, moderate etc.

Response 1: Thank you for your correction, we changed accordingly and added the Ethics file number and Æž2 descriptor in methodology section.

Point 2: Overall the results are well-written and showed

Overall, the discussion is well-written and incorporates relevant literature.

These sections are well designed and well-written.

Response 2: Thank you for the comments

Reviewer 4 Report

The paper is an intersting study in providing association between some anthropometric characteristics and performance in very young water polo players. The research's design is appropriate. The findings are interesting for describing the model of performace in youg water polo players.  However, the english need an extensive review (it miss some articles, the phrases contructions are somewhere inverted and so on. After a deep revision of english the paper can be accepted. It is an interesting descriptive study searching for association between some technical skills and anthropometrics and can be of inteterest for water polo players coaches.

Author Response

The paper is an interesting study in providing association between some anthropometric characteristics and performance in very young water polo players. The research's design is appropriate. The findings are interesting for describing the model of performance in young water polo players. It is an interesting descriptive study searching for association between some technical skills and anthropometrics and can be of interest for water polo players coaches.

Response: Thank you for the comments. We would like to acknowledge for the time expended on reviewing our manuscript.

Point 1: However, the English need an extensive review (it miss some articles, the phrases constructions are somewhere inverted and so on. After a deep revision of English, the paper can be accepted.

Response 2: Thank you for your detailed review of our manuscript. We have made a re-analysis of the English language, which was significantly improved.

Reviewer 5 Report

Youth water polo performance determinants: The INEX study

The aim of the study was to provide basic anthropometrics and general physical fitness attributes as well as sport-specific performance tests data on youth-level water-polo athletes , from 12 to 14 y old, from Serbia.

Major concerns:

- the lacked of control group – youths of the same age. Because no control group, we can be sure why cohort 2 group showed better anthropometrics and performance  than cohort 3.

- The present participants of the study training and competitive status should be further described so that the readers can appreciate whether these youth individuals are of highly trained or well-trained or merely recreational-level players. Authors should clearly detail their years and experience in training and competing as well as type of training they currently undergoing, and any other relevant info.

- Table 1 to Table 3. All tables were missing the post-hoc paired t-test after the ANOVA test – to determine which pair of the cohort age-group showed differences between each other. Similarly, it is more useful to show the effects size differences between paired (or 2) groups rather among the 3 groups.

- Line 88-92. Please provide a reference to this method of determining your sample’s peak heigh velocity and also the reason (or validity) for using this method/calculation with your study’s population.

Minor issues

- throughout the manuscript. Use “stature” instead of “height” and “body mass” instead of “weight”.

- - Also, I would recommend that the authors to conduct a paired corelations analysis (on the combined sample) between the general test, say for example, the YOYO IR test with the more specific in-water aerobic fitness – if there is a strong positive correlation here, then perhaps the general test of fitness can be useful to help detect individuals for the sport of water-polo from such “land-based field tests”. Authors should try on all the other tests such as between medicine ball throw and swim sprint 10 m, etc.

- “M±Sd” should be written as “mean±SD”.

- Line 13. Delete the mean age numbers because it does not add value to the abstract because you are analysing the data in the three age-groups.

- Line 50. “fine” is not appropriate. Use “optimal” or “specific”.

- Line 76-77. More details of INEX are required here. For example, what’s the purpose of this project what the study is measuring and so. OR provide a relevant reference here.

- Line 109. “Players’ sport-specific or water-polo physiological …..”. Add the underlined.

- Line 116. Delete “It were conducted two trails,” and changed to “Two trials were conducted. …..”.

- Line 149 to 155. The TEM of all the measures and tests should be written in the methods section when detailing each measure or test.

- Line 165. Add the word “younger” before “counterparts”.

- Line 203 to 210. The information that you have collected here does not seem to be relevant to the aims of the study. Please clarify the useful and benefits of these information acquired to the purpose of your study.

- Table 4. There seems to be misalignment of the variables with the specific equations. Please ensure that the alignment is clearer and have space between rows to reflect clearly.

- Line 240. “…. players of similar age than our cohort 1 and 2 ….”; I think “than” is wrong word; please use “to”.

- Line 270. Add “from physical fatigue” after the word “match”.

- Please highlight the limitations of the study – the small sample size for Cohort 3.

- Line 321-327. Authors need to be more specific with their written conclusions. Line 323, authors should specifically mention which measure(s) of anthropometrics and which general motor performance test(s) that are closely associated or related to which specific motor abilities performances in youth water polo.

Author Response

The aim of the study was to provide basic anthropometrics and general physical fitness attributes as well as sport-specific performance tests data on youth-level water-polo athletes , from 12 to 14 y old, from Serbia.

Response: We would like to acknowledge for the time expended on reviewing our manuscript. However, we would like to address that the athletes were not from Serbia, that was used to compare with our study.  

Point 1: Major concerns:

- the lacked of control group – youths of the same age. Because no control group, we can be sure why cohort 2 group showed better anthropometrics and performance  than cohort 3.

Response 1: We understand your opinion, but the aim of the study was to address the “potential association of the assessed specific motor abilities with anthropometric characteristics and general motor performance scores” and therefore we believed that in this particularly study the lack of control group is not an issue.

- The present participants of the study training and competitive status should be further described so that the readers can appreciate whether these youth individuals are of highly trained or well-trained or merely recreational-level players. Authors should clearly detail their years and experience in training and competing as well as type of training they currently undergoing, and any other relevant info.

Response 2: The competitive years were added as proposed.

- Table 1 to Table 3. All tables were missing the post-hoc paired t-test after the ANOVA test – to determine which pair of the cohort age-group showed differences between each other. Similarly, it is more useful to show the effects size differences between paired (or 2) groups rather among the 3 groups.

Response 3: Thank you for pointing it out. The post-hoc comparisons were added in the tables as proposed.

- Line 88-92. Please provide a reference to this method of determining your sample’s peak heigh velocity and also the reason (or validity) for using this method/calculation with your study’s population.

Response 4: Thank you for the comment. We added the missing reference.

Point 2: Minor issues

- throughout the manuscript. Use “stature” instead of “height” and “body mass” instead of “weight”.

Response 5: We have changed accordingly with your suggestion.

-  Also, I would recommend that the authors to conduct a paired corelations analysis (on the combined sample) between the general test, say for example, the YOYO IR test with the more specific in-water aerobic fitness – if there is a strong positive correlation here, then perhaps the general test of fitness can be useful to help detect individuals for the sport of water-polo from such “land-based field tests”. Authors should try on all the other tests such as between medicine ball throw and swim sprint 10 m, etc.

Response 6: We respect the reviewer’s suggestion, and the concept is interesting. However, since these tests assess different constructs, eventual correlations could be spurious. Indeed, even strong correlations could be spurious – since multiple correlations would be calculated, the likelihood of finding a “significant” correlation by chance would also increase and could therefore mislead the analysis. Moreover, that line of argumentation – although valid and relevant – would deviate from the main focus of this research. Finally, the association models in table 4 suggest lack of collinearity between the general tests and their best-matched / corresponding specific tests.

- “M±Sd” should be written as “mean±SD”.

- Line 13. Delete the mean age numbers because it does not add value to the abstract because you are analysing the data in the three age-groups.

- Line 50. “fine” is not appropriate. Use “optimal” or “specific”.

- Line 76-77. More details of INEX are required here. For example, what’s the purpose of this project what the study is measuring and so. OR provide a relevant reference here.

- Line 109. “Players’ sport-specific or water-polo physiological …..”. Add the underlined.

- Line 116. Delete “It were conducted two trails,” and changed to “Two trials were conducted. …..”.

- Line 165. Add the word “younger” before “counterparts”.

- Table 4. There seems to be misalignment of the variables with the specific equations. Please ensure that the alignment is clearer and have space between rows to reflect clearly.

- Line 240. “…. players of similar age than our cohort 1 and 2 ….”; I think “than” is wrong word; please use “to”.

- Line 270. Add “from physical fatigue” after the word “match”.

- Please highlight the limitations of the study – the small sample size for Cohort 3.

- Line 149 to 155. The TEM of all the measures and tests should be written in the methods section when detailing each measure or test.

Response 7: Changed/added accordingly with your suggestion.

- Line 203 to 210. The information that you have collected here does not seem to be relevant to the aims of the study. Please clarify the useful and benefits of these information acquired to the purpose of your study.

Response 8: We understand your opinion, but we believe that this information will help to understand the background of athletes training regarding quality of the training.

- Line 321-327. Authors need to be more specific with their written conclusions. Line 323, authors should specifically mention which measure(s) of anthropometrics and which general motor performance test(s) that are closely associated or related to which specific motor abilities performances in youth water polo.

Response 9: The reviewer is correct, and we apologize for the lapse. In-water vertical jump and 10 m sprint were both associated with upper limbs explosive power, as well as with the combination of body mass and arm span. Change of direction was associated with arm span, while aerobic fitness was associated with arm span, and independently with upper limbs explosive power.

Reviewer 6 Report

Review of the manuscript entitled: Deranged Haematological Profile and Dyslipidaemia in Diabetes Induced Nephropathy  The manuscript submitted is appropriate to the subject matter and scientific rigor. The authors raised a very current issue at work, which is not only interesting from a scientific but also a practical point of view. Some remarks improving the quality of future research. and suggested changes and comments to the submitted manuscript in order to improve the quality of the planned research and future publications below:

  1. Maybe it is worth extending the conclusions. For example in which the most important skills are necessary to achieve success with players water polo? What should coaches pay attention to while developing training programs for them?
  2. Would you please add the DOI and Issue to some references. accordance with the journal's guidelines.
  3. Would you please check your references number: 27, 34 and 51.

More comments you can find in your manuscript which i send to you in attachment.

Author Response

Response: Thank you for the time spent analysing our study. We have tried to follow each comment but sometimes we had difficulties to understand exactly what you suggested.

Commented [u1]: Would you please check these references. In articulates the authors write about biomechanical determinants. The conclusion was” would you please check this reference . The conclusion of this article was : „Conclusions: On the one hand, our findings demonstrate the great capability of sport specific technical skills assessments to discriminate different performance levels and predict future performance in TID activities. On the other hand, this review highlights the focus on 'outcome-related' and 'experimental' methods in specific populations and, consequently, the limited knowledge in other areas. Here, the application of 'technique-related' and 'competition' methods appears promising for adding new knowledge, especially in the light of technological advances.

Response: We wanted to change accordingly but we did not understand exactly what you meant.

Commented [u2]: Maybe it is also worth here which training sessions were combined, for example "..... HIIT and strength training together with specific water polo training performed concurrently"

Response: We considered your suggestion.

Commented [u3]: The research was conducted here among football players not water polo players the Conclusion reference number [27] was: We conclude that the test battery used may be useful in establishing baseline reference data for young players being selected for specialized development programs ... " I think that authors need other research carried out in the aquatic environment among water polo players

Commented [u6]: Maybe some reference form cooch water polo not only ice hockey, soccer, and baseball ???

Responses 3 and 6: Following your suggestion, we have changed the references.

Commented [u4]: The research was done in young elite skiers (109 females, 138 males) I think it should be written here and the study was conducted in a group water polo and additionally refer to the references regarding the discussed topic among

Commented [u5]: The research was done in basketball coaches I think it should be written here and the study was conducted in a group water polo and additionally refer to the references regarding the discussed topic among water polo players

Commented [u7]: note as above

Response: We understand your point of view but there are not studies available in water polo on this topic. This is the reason why we have added studies in other sports.

Commented [u8]: only one? Would you please explain it or in 2 or 3 more references. please check more publications of the authors of the article he refers to. Are you sure they haven't published anything else

Response: We understand your concern, but this statement is based on our literature review. Yes, we are sure that no other publication in peer reviewed journals with JCR impact factor.

Commented [u9]: why here the authors made a reference to the literature [12-14] at this point, please delete it

Response: These are not references but minimum and maximum ages. We have rewritten aiming for higher clarity.

Commented [u10]: In the description of the results, the authors provide numbers (e.g. in Table 1- The older players were heavier (F2.92 = 4.25)) that are not in the tables, explain it or describe the results differently or enter these numbers in the table1

Commented [u11]: note as above

Commented [u12]: note as above

Response: We have changed accordingly with your suggestion. We think it is better described now.

Commented [u13]: we can read in „In conclusion, incremental increases in fatigue differentially influenced decision making (improved) relative to the technical performance (declined), accuracy and speed of the ball (unchanged) of a water polo goal shot. „ Would you please check this reference

Response: we have checked de reference and maintain it since it is accurate.

Commented [u14]: Maybe it is worth extending the discussion with information from which of the respondents: anthropometric characteristics or which motor skills are essential to success in matches and the overall development of players over the years? Which trainers should pay special attention to in the training process

Response: We have tried to go directly to the point and to be concise in the conclusions section. However, many implications for training were include in each paragraph of the discussion section. We hope that these pleases you.

Commented [u15 – u63]: References

Response: The reviewer is correct and we apologize for the gap. So, we have changed/added accordingly with your suggestion, with exception of the journal issue that, in accordance with journal references guidelines, should not be included.

Round 2

Reviewer 5 Report

Just putting down that the players has 2 years of competitive experience is not sufficient. The training underwent should be well-described. For example, how hours of training per week does the groups underwent over the last 3 months prior to when the data was collected. Training frequency and duration should be reported. The types of training should also be well-described. Only then will the reader appreciate the "status" of this group of athletes. 

Author Response

Just putting down that the players has 2 years of competitive experience is not sufficient. The training underwent should be well-described. For example, how hours of training per week does the groups underwent over the last 3 months prior to when the data was collected. Training frequency and duration should be reported. The types of training should also be well-described. Only then will the reader appreciate the "status" of this group of athletes. 

Response: Thank you for your suggestion, we changed accordingly.